# A Novel Strategy to Identify Endolysins with Lytic Activity against Methicillin-Resistant *Staphylococcus aureus*

**DOI:** 10.3390/ijms24065772

**Published:** 2023-03-17

**Authors:** Hanbeen Kim, Jakyeom Seo

**Affiliations:** Department of Animal Science, Life and Industry Convergence Research Institute, Pusan National University, Miryang 50463, Republic of Korea; khb3850@pusan.ac.kr

**Keywords:** prophage, endolysin, *Staphylococcus aureus*, methicillin-resistant *Staphylococcus aureus*, antibiotic resistance, bovine mastitis

## Abstract

The increasing prevalence of methicillin-resistant *Staphylococcus aureus* (MRSA) in the dairy industry has become a fundamental concern. Endolysins are bacteriophage-derived peptidoglycan hydrolases that induce the rapid lysis of host bacteria. Herein, we evaluated the lytic activity of endolysin candidates against *S. aureus* and MRSA. To identify endolysins, we used a bioinformatical strategy with the following steps: (1) retrieval of genetic information, (2) annotation, (3) selection of MRSA, (4) selection of endolysin candidates, and (5) evaluation of protein solubility. We then characterized the endolysin candidates under various conditions. Approximately 67% of *S. aureus* was detected as MRSA, and 114 putative endolysins were found. These 114 putative endolysins were divided into three groups based on their combinations of conserved domains. Considering protein solubility, we selected putative endolysins 117 and 177. Putative endolysin 117 was the only successfully overexpressed endolysin, and it was renamed LyJH1892. LyJH1892 showed potent lytic activity against both methicillin-susceptible *S. aureus* and MRSA and showed broad lytic activity against coagulase-negative staphylococci. In conclusion, this study demonstrates a rapid strategy for the development of endolysin against MRSA. This strategy could also be used to combat other antibiotic-resistant bacteria.

## 1. Introduction

*Staphylococcus aureus* is a Gram-positive bacterium that forms part of the normal microbiome residing on the skin of animals and humans [1,2]. *Staphylococcus aureus* is considered an opportunistic pathogen and is involved in a variety of infectious diseases, such as skin infections, food poisoning, toxic shock syndrome, and endocarditis [3,4]. In addition, the treatment of *S. aureus* infections is challenging because of the emergence of antibiotic-resistant *S. aureus* strains, such as methicillin-resistant *S. aureus* (MRSA), vancomycin-resistant *S. aureus*, and multidrug-resistant *S. aureus* [5,6].

In the dairy industry, *S. aureus* is considered one of the most concerning and contagious pathogens, as it can cause bovine mastitis. This disease triggers an inflammatory response in the udder tissue, causing enormous economic losses to dairy farms [7]. A recent study reported that MRSA is a critical risk factor for bovine mastitis, representing 12.2% of all mastitis cases [8]. In addition, evidence has emerged suggesting that MRSA can be directly transmitted from cows to humans working in the dairy industry [9]. Considering the poor efficacy of β-lactam antibiotics against MRSA, alternative strategies are required to combat this pathogen in dairy farms.

Endolysins are bacteriophage-derived peptidoglycan hydrolases and are utilized in the final stage of the bacteriophage replication cycle to release newly produced progeny from the host bacterial cell by breaking down the peptidoglycan layer, thereby inducing the lysis of host bacteria. Endolysins active against Gram-positive bacteria have generally shown multiple combinations of conserved domains, especially the N-terminal enzymatically active domain (EAD) and the C-terminal cell wall-binding domain (CBD) [10]. In general, a CBD of endolysin determines the spectrum of lytic activity by recognizing and binding to the peptidoglycan layer [10]. The use of endolysins as alternatives to antibiotics has been highlighted because they exhibit high specificity against host bacteria and low resistance rates while eliciting and targeting lysis through rapid reactions [11].

Many efforts have been made to develop anti-staphylococcal agents, such as bacteriocins [12,13,14] and endolysins [15,16,17], and a recent review highlighted their potential as therapeutic agents against antibiotic-resistant *S. aureus*. A general strategy to develop endolysins starts from the isolation of bacteriophages with lytic activity against target bacteria; however, the isolation of specific bacteriophages with lytic activity against a pathogen of interest is laborious and time-consuming. Recent studies have suggested a new strategy to develop a recombinant endolysin using genomic information from prophages [18,19,20]. However, the vast amount of genetic information in the database makes it challenging to isolate specific endolysins that can effectively lyse target bacteria.

In this study, we hypothesized that identified endolysin with proper CBD from MRSA genomes might have lytic activity against *S. aureus* and MRSA. Therefore, the objective of this study was to establish a strategy for identifying and selecting endolysin candidates from MRSA genome information using bioinformatical approaches and develop new endolysins with lytic activity against *S. aureus* and MRSA.

## 2. Results

### 2.1. Identification of Putative Endolysins against MRSA

In total, 707 complete genomes of *S. aureus* were retrieved from the NCBI database (retrieval date: 6 February 2020), and 474 of the complete *S. aureus* genomes exhibited PBP2a (Figure 1a and Appendix A). Among the MRSA prophage sources, 1907 sequences were detected as endolysins; however, only 272 sequences remained for further analysis after removing sequences containing duplicate and incomplete conserved domains (Figure 1b and Appendix A). Among these 272 putative endolysin sequences, 42.3% contained a single domain (*n* = 115) and the rest exhibited multi-conserved domain combinations (*n* = 157; Figure 1c). The most frequent domain among the putative endolysins with a single domain was the LytD domain (approximately 52.2%, accession: COG4193), followed by the cysteine, histidine-dependent amidohydrolase/peptidase (CHAP) domain (approximately 19.1%, accession: pfam05257), and the SH3b domain (approximately 13.9%, accession: smart00287; Figure 1d). The most frequently observed multi-conserved domain combination was (CHAP (Accession: pfam05257) + Amidase_2 (Accession: smart00644) + SH3_5 (Accession: pfam08460)), followed by (CHAP (Accession: pfam05257) + Amidase_3 (Accession: pfam01520) + SH3b (Accession: smart00287)) and (CHAP (Accession: pfam05257) + LytD (Accession: COG4193)) (Figure 1e). Interestingly, among the 157 putative endolysin sequences with multi-conserved domain combinations, only 114 putative endolysin sequences showed both an EAD and a CBD, indicating general characteristics for endolysins targeting Gram-positive bacteria (Appendix A).

### 2.2. Sequence Comparison among Putative Endolysins Targeting MRSA

The 114 putative endolysins with both an EAD and a CBD were divided into three groups based on their shared conserved domain combinations (groups 1, 2, and 3). The overall distributions of the 114 putative endolysins, as determined by the multi-alignment results and identified phylogenetic distances, were visualized using a cladogram plot (Figure 2a). Among the three groups, putative endolysins belonging to groups 1 and 2 showed not only close phylogenetic relationships (belonging to the same clade) but also high sequence identity (78.2%, data not shown), sharing CHAP (pfam05257) and Amidase_2 (smart00644) domains (Figure 2b). However, putative endolysins from group 3 belonged to different clades from those of groups 1 and 2 (Figure 2a) and shared low sequence identity (20.6%, data not shown). Interestingly, putative endolysins belonging to group 3 were more conserved within the group (87.8%) than those belonging to group 1 (80.5%; data not shown).

According to the multi-alignment of each conserved domain from the putative endolysins in group 1, CHAP and Amidase_2 domains were highly conserved (CHAP domain, sequence identity: 83.5%; Amidase_2 domain, sequence identity: 82.4%), whereas the SH3_5 domain exhibited slightly lower sequence similarity (sequence identity: 75.8%) (Appendix A). In group 2, both Amidase_3 and SH3_b domains exhibited higher sequence identity than any of the domains in group 1 (Amidase_3 domain, sequence identity: 95.1%; SH3_b domain, sequence identity: 88.4%) (Appendix A). Figure 3 shows overall comparisons of each conserved domain sharing a similar name and/or function from the 114 putative endolysins with an EAD and a CBD. Interestingly, the Amidase-like domain (Figure 3a; Amidase_2 and Amidase_3) and SH3-like domain (Figure 3b; SH3_5, SH3 superfamily, and SH3b) showed extremely low sequence identity (amidase-like domain, 12.7%; SH3-like domain, 22.9%), whereas the CHAP domain exhibited high sequence identity (Figure 3a; 82.4%).

### 2.3. Comparison of Predicted Recombinant Protein Solubility

Recombinant protein solubility was predicted based on three different bioinformatics tools (Soluprot, Protein-Sol, and SKADE), and results are presented in Table 1 and Appendix A. Because different prediction tools yielded different ranges of predicted solubility scores, we first normalized the predicted solubility scores and then summed them to compare the predicted solubilities. Among the top 10 putative endolysins, 9 belonged to group 1. The top putative endolysin, number 177, belonged to group 3. Based on the predicted solubility, we selected two putative endolysins, putative endolysin 177 (group 3) and putative endolysin 117 (group 1).

### 2.4. Overexpression and Structural Analysis of Selected Endolysins

Of the two endolysin candidates, only recombinant putative endolysin 117 was successfully overexpressed and purified in its soluble form; we renamed it LyJH1892. LyJH1892 comprised 481 amino acids, and its molecular weight was predicted to be 53.8 kDa. The major band of purified soluble LyJH1892 confirmed a molecular mass close to 53.8 kDa upon sodium dodecyl sulfate-polyacrylamide gel electrophoresis (SDS-PAGE; Figure 4a). The conserved domains of LyJH1892 included three distinct domain architectures (Figure 4b): (1) N-terminal CHAP domain (Accession: pfam05257, e-value = 2.21 × 10^−8^, and bit-score = 51.3), (2) Amidase_2 domain (Accession: smart00644, e-value = 5.22 × 10^−24^, and bit-score = 96.7), and 3) SH3_5 domain (Accession: pfam08460, e-value = 6.20 × 10^−22^, and bit-score = 88.9) (Appendix A). The three-dimensional (3D) structure of LyJH1892 was predicted using AlphaFold2 and visualized in ribbon form and Connolly surface form using ChimeraX 1.3 (Figure 4c,d). The overall secondary structure of LyJH1892 comprised alpha-helices (22.7%), beta sheets (21.8%), and random coils (55.5%). The CHAP domain contained 27.7% alpha helices, 22.9% beta sheets, and 49.4% random coils, and the Amidase_2 domain showed a secondary structure similar to that of the CHAP domain (Amidase_2 domain: alpha helices, 21.6%; beta-sheet, 18.4%; and random coil, 60.0%). Unlike these domains, the SH3_5 domain consisted of 51.5% beta sheets and 48.8% random coils.

### 2.5. Antibiotic Resistance Profiles of S. aureus Strains

Among the four strains of *S. aureus*, *S. aureus* (NCCP 16830) was determined as methicillin-susceptible *S. aureus* (MSSA), and the others were determined as MRSA based on the diameter of the clear zone by cefoxitin and oxacillin (Table 2 and Appendix A). All *S. aureus* strains demonstrated susceptibility to chloramphenicol while exhibiting resistance to penicillin. In addition, all MRSA strains showed similar antibiotic resistance profiles (Table 2 and Appendix A).

### 2.6. Characterization of Recombinant LyJH1892

We assessed the lytic activity of LyJH1892 against *S. aureus* (NCCP 16830) at various pH values, temperatures, NaCl concentrations, and metal ion additions to identify the optimal lytic conditions for LyJH1892. The lytic activity of LyJH1892 was significantly higher at pH 6.0 and 9.0 than at 10.0 (*p* = 0.0089). Additionally, the lytic activity was highly stable at pH 6.0–9.0 (Figure 5a). The optimal temperature for LyJH1892 activity was 25 °C, and moderate activity (above 50%) was observed at 4–37 °C; however, the lytic activity of LyJH1892 significantly decreased above 50 °C (Figure 5b, *p* = 0.0065). The highest lytic activity of LyJH1892 was detected at a NaCl concentration of 0 mM (Figure 5c, *p* = 0.0054). Moderate activity (above 60%) was observed at a NaCl concentration of 0–62.5 mM, whereas low lytic activity (approximately 30% or less) was noted beyond a NaCl concentration of 125 mM. In the metal ion test, the addition of 5 mM ethylenediaminetetraacetic acid (EDTA) to LyJH1892 significantly decreased its lytic activity relative to that of pure LyJH1892 (Figure 5d, *p* = 0.0065). The addition of 10 mM Ca^2+^, Mg^2+^, or Mn^2+^ to EDTA-treated LyJH1892 restored lytic activity to above 65% but did not fully recover the lytic activity of LyJH1892 (Figure 5d).

### 2.7. Optimal Lytic Activity and Lytic Spectrum of LyJH1892

Under optimal conditions (pH 9.0, 25 °C, and no addition of NaCl or metal ions), a dose–response test of LyJH1892 against *S. aureus* (NCCP 16830) was performed. The lytic activity of LyJH1892 increased in a dose-dependent manner, and the highest dosage (100 μg/mL) lysed approximately 85% of *S. aureus* cells (Figure 6a). To evaluate the lytic spectrum of LyJH1892, we further analyzed its lytic activity under optimal conditions against three strains of MRSA (KVCC-BA0500624, NCCP 14754, and NCCP 14567), and seven species of coagulase-negative staphylococci (CNS) (*S*. *epidermidis* (KVCC-BA0001452), *S*. *chromogenes* (KCTC 3579), *S*. *hyicus* (KVCC-BA0001860), *S*. *haemolyticus* (NCCP 14693), *S*. *warneri* (NCCP 16234), *S*. *simulans* (NCCP 16236), and *S*. *xylosus* (KCCM 40887)) and *S*. *agalactiae* (NCCP 14729) (Table 3). LyJH1892 showed potent lytic activity against MRSA strains (Figure 6b). Among the CNS species, the highest lytic activity was observed against *S. epidermidis* (relative lytic activity, approximately 145%). High lytic activity was also observed against *S. chromogenes*, *S. hyicus*, and *S. haemolyticus* (relative lytic activity > 70%; Figure 6b). However, LyJH1892 showed low lytic activity against two CNS species (*S. warneri* and *S. simulans*) and *S. agalactiae* (Figure 6b).

## 3. Discussion

In this study, we introduced a strategy to develop putative endolysins targeting MRSA based on genome information and bioinformatics tools, rather than the traditional bacteriophage screening approach. We identified 272 putative endolysin candidates from 474 MRSA genome sequences and successfully developed and characterized a new endolysin, LyJH1892, that shows high lytic activity against *S. aureus* and MRSA.

Biosynthesis of the peptidoglycan layer, which is composed of disaccharide *N*-acetylglucosamine-*N*-acetylmuramic acid with peptide stems, is a fundamental process for bacterial survival during growth and division [21]. The crosslinking reaction between *N*-acetylmuramic acid and peptide stems is mediated by transpeptidases, also known as PBPs [22]. In general, the native PBPs of *S. aureus* include PBP1, PBP2, PBP3, and PBP4 [23]; however, MRSA possesses a unique PBP called PBP2a [24,25]. *β*-lactam antibiotics bind to the native PBPs of *S. aureus* and block the reconstruction of the peptidoglycan layer, but MRSA can survive because PBP2a has a low affinity for *β*-lactam antibiotics [23]. A previous study suggested that the *mec*A gene, which encodes PBP2a, could be a useful marker for methicillin resistance in *S. aureus* [26]. Therefore, we used PBP2a as a computational marker to identify MRSA among the whole genome of *S. aureus* listed in the NCBI database. We found that approximately 67% of *S. aureus* strains were identified as MRSA among 707 whole genomes. A meta-analysis of 37 studies reported that the pooled prevalence of MRSA among *S. aureus* in China was 15% [27]. In Europe, the proportion of MRSA among *S. aureus* was 19% in 2008 [28]. Therefore, the prevalence of MRSA in our study was significantly higher than that reported in previous studies. However, several studies have reported a similar or even higher prevalence of MRSA than that in our study: 72% in Eritrea [29], 80% in Peru [30], and 90% in Colombia [31]. According to the existing literature on the prevalence of MRSA among *S. aureus* strains, several factors can be attributed to this variation, including geographical variation, study design, MRSA detection method, and study population [27,32]. Another possible explanation for the high prevalence of MRSA in our study could be that the data on *S. aureus* genome sequences available in the NCBI database may be subject to bias due to a heightened focus on antibiotic resistance concerns.

In the present study, we identified 272 putative endolysin sequences from MRSA genomes, with many variations in amino acid sequences and conserved domains. The CHAP domain is known as an EAD with two different peptidoglycan catalytic activities: *N*-acetyl muramyl-L-alanine amidase and endopeptidase. These split crosslinked peptide stems, especially D-alanine-glycine [33]. A previous study suggested that the CHAP domain is predominantly found in phage endolysins that infect Gram-positive bacteria, especially *Staphylococcus* spp. [34]. SH3 domains were first identified in eukaryotes and are generally involved in cell-to-cell communication and intracellular signaling from the cell membrane to the nucleus by binding to proline-rich polypeptides [35,36]. However, hundreds of SH3 domains, which include SH3b, SH3-3, or SH3-5 types, were also found in bacteria and bacteriophage proteins, such as autolysin and endolysin structures [34,37]. Many studies have verified the role of bacterial SH3 domains as binding modules found in endolysins targeting *S. aureus* [38], several *Bacillus* species [39], and *Listeria monocytogenes* [40]. In addition, the host specificity or range of endolysins is determined by the type of C-terminal CBD [41,42,43]. Therefore, we focused on the 114 putative endolysins (approximately 72.6%) with multi-conserved domain combinations possessing both an EAD and a CBD. We found that 53.5% of endolysins had a CHAP domain as an EAD, and all endolysins exhibited SH3-like CBDs. In a previous study, Becker et al. [36] investigated *Staphylococcus* phage proteins containing SH3 domains (SH3b and SH3_5) and divided SH3-containing endolysins into five groups based on their sequence similarity (>90%) (group 1, CHAP—Amidase_3 and SH3b domain; group 2 and 3, CHAP, Amidase_2, and SH3b domains; group 4, CHAP and SH3b domains; and group 5, glycyl-glycine endopeptidase, and SH3b domain). Recently, Chang and Ryu [17] reported that 98 *Staphylococcus* endolysins identified from phage sequences in the NCBI database were divided into six groups based on their conserved domain architecture (group 1, CHAP + unknown CBD; group 2, CHAP + SH3 domain; group 3, CHAP + Amidase_2 + and SH3 domain; group 4, CHAP, Amidase_3, and SH3 domains; group 5, CHAP, Amidase_3, and CBD (unknown); and group 6, Peptidase_M23, Amidase_2, and SH3 domains). The majority of endolysins from the phage source represented a combination of CHAP, Amidase, and SH3-like domains (Becker et al. [36], 89.5%; Chang and Ryu [17], 68.4%), though some endolysins did not have a CHAP domain (Becker et al. [36], 3.5%; Chang and Ryu [17], 1.0%). Interestingly, in the present study, a different type of domain architecture comprising Amidase_3 and SH3b domains (group 3) was observed, occupying 46.5% of the putative endolysins. Based on the amino acid sequences of these endolysins, group 3 showed the highest sequence similarity (Amidase_3 domain, 95.1%; SH3_b domain, 88.4%) compared to groups 1 and 2. In this study, CHAP domains from groups 1 and 2 were highly conserved (overall CHAP domains, 82.4%); however, Becker et al. [36] reported low conservation of CHAP domains (less than 50% identity). In addition, they also reported a higher sequence similarity between Amidase_2 and Amidase_3 (53% identity) than that in our study (overall sequence similarity in amidase-like domains, 12.7%). In the present study, we found that the SH3-like domains contained 61–68 amino acids and exhibited low sequence similarities with 10 perfectly conserved amino acid positions (glycine, 3; tyrosine, 3; valine, 1; arginine, 1; cysteine, 1; and tryptophan, 1). However, Becker et al. [36] reported that alignment of the 11 SH3_5 domains from endolysins identified from *Staphylococcus* phages showed six perfectly conserved amino acid positions (glycine, 2; glutamic acid, 1; arginine, 1; tyrosine, 1; tryptophan, 1). The endolysins identified in this study showed low sequence similarity and different positions of conserved amino acids, mainly in SH3-like domains compared to endolysins from *Staphylococcus* phages. Therefore, we speculated that the newly identified 114 putative endolysins are derived from different ancestors than the endolysins described in previous studies.

The production of heterologous proteins in *E. coli* systems is often limited by the low yield of soluble proteins or the production of insoluble proteins [44]. Although several empirical processes can be used to optimize the production of heterologous proteins in soluble form (e.g., promoter design, codon optimization, and culture condition manipulation) [45], these processes are generally laborious and time-consuming. Bioinformatic tools can be used to predict protein solubility via sequence-based and machine-learning-based methods [46,47]. In the present study, we evaluated the predicted protein solubility using three different bioinformatic tools, including a sequence-based method (Protein-Sol) [48] and two machine-learning-based methods (SoluProt and SKADE) [49,50]. Machine-learning-based methods yielded generally higher solubility values for putative endolysins belonging to group 1 than for those in group 3, whereas the opposite trend was observed using the sequence-based method. Based on the integrated protein solubility indices, we attempted to overexpress heterologous endolysins (putative endolysins 177 and 117). Despite the high solubility scores, only putative endolysin 117 (LyJH1892) was successfully produced in its soluble form. Overall, we propose that the use of machine-learning-based solubility prediction tools could represent a useful strategy to identify endolysins among putative candidates. However, due to limited data, further research should be conducted to verify more accurate standards for the selection of endolysin candidates using solubility prediction tools.

The increasing appearance of antibiotic-resistant *S. aureus* has become a serious problem because this pathogen causes bovine mastitis and lethal infections in livestock handlers [51,52]. In this study, we evaluated the antibiotic susceptibility test against four strains of *S. aureus*, and only one strain was susceptible to cefoxitin and oxacillin, which are commonly recommended antibiotics for methicillin resistance screening [53]. In addition, three strains determined as MRSA showed similar antibiotic resistance profiles except for clindamycin. In this study, we successfully characterized an endolysin, LyJH1892, with lytic activity against *S. aureus* and MRSA. Even though all of the *S. aureus* strains used in this study exhibited susceptibility to chloramphenicol at lower dosages compared to LyJH1892, several studies have reported the prevalence of *S. aureus* strains resistant to chloramphenicol [54,55,56], which suggests that antibiotics are not free from resistance problems. Further, phage therapy is a promising choice as an alternative to antibiotics because it shares advantages with endolysins, including rapid lytic activity and high specificity [57]. Unlike endolysins, several studies reported problems: (1) narrower specificity than endolysins, (2) a certain density of host bacteria as a requirement for phage proliferation, and (3) the emergence of phage resistance in bacteria [58,59]. In this aspect, our strategy is feasible for the development of endolysins against antibiotic-resistant *S. aureus* strains.

In addition, CNSs are considered an emerging risk factor for bovine mastitis, inducing intramammary infections and negatively impacting udder health [7,60,61,62,63]. In general, the staphylococcal peptidoglycan belongs to the A3α subgroup and is crosslinked by pentaglycine bridges [64]. Lysostaphin is a glycyl-glycine endopeptidase secreted by *S. simulans* with a known bactericidal effect against *S. aureus* [13]. Like lysostaphin, ALE-1 is a glycyl-glycine endopeptidase secreted by *Staphylococcus capitis* EPK1 that also exhibits lytic activity against *S. aureus* and other *Staphylococcus* species [12,14]. Both peptidoglycan hydrolases are known to have SH3-like domains (SH3b domain) that share a high degree of homology. Additionally, previous studies have shown that the SH3b domains of these enzymes require intact penta-glycine cross-bridges for binding [65,66]. However, LyJH1892 showed potent lytic activity against only four of the seven CNS species, including *S. chromogenes*, *S. haemolyticus*, *S. epidermidis*, and *S. hyicus*. These unexpected specificity spectrum results can be explained by several possible reasons. Previous studies have demonstrated that truncated endolysins consisting of only EAD also lyse staphylococcal species, indicating that the binding spectrum of endolysin is not determined by the characteristics of the CBD alone [67,68]. In addition, Lu et al. [65] reported that the outside region of the SH3b domain in ALE-1 is essential for the full recovery of binding affinity and demonstrated that the composition of penta-amino acids in the peptidoglycan layer affects the lytic spectrum of the SH3b domain of ALE-1, showing a strong preference for penta-glycine. Furthermore, variations exist in interpeptide bridges among staphylococci and streptococci belonging to peptidoglycan subgroup A3α [64,69]. This could be one of the reasons for the unexpected result from the present lytic spectrum. However, based on the potent lytic activity of LyJH1892 against MSSA, MRSA, and several important pathogenic CNSs, LyJH1892 could be a useful biocontrol agent for bovine mastitis.

## 4. Materials and Methods

### 4.1. Bacterial Strains and Growth Conditions

The bacterial strains used in the present study are listed in Table 3. *Escherichia coli* cloning strain DH5α and expression strain BL21 (DE3) were aerobically grown at 37 °C in Luria-Bertani (LB) broth. *Staphylococcus aureus* (NCCP 16830), MRSA (NCCP 14567), MRSA (NCCP 14754), *S. haemolyticus* (NCCP 14693), *S. simulans* (NCCP 16236), *S. warneri* (NCCP 16234), and *Streptococcus agalactiae* (NCCP 14729) were obtained from the National Culture Collection for Pathogens (Cheongju, Republic of Korea). MRSA (KVCC-BA0500624), *S. hyicus* (KVCC-BA0001860), and *S. epidermidis* (KVCC-BA0001452) were obtained from the Korea Veterinary Culture Collection (Gimcheon, Republic of Korea). *Staphylococcus chromogenes* (KCTC 3579) and *S. xylosus* (KCCM 40887) were obtained from the Korea Collection for Type Cultures (Jeongeup, Republic of Korea) and the Korean Culture Center of Microorganisms (Seoul, Republic of Korea), respectively. All staphylococcal species and *S*. *agalactiae* were grown aerobically at 37 °C in brain heart infusion (BHI) broth.

### 4.2. Antibiotic Susceptibility Test

The antibiotic resistance profiles of *S. aureus* strains (NCCP 16830, NCCP 14567, NCCP 14754, and KVCC-BA0500624) were investigated using disc diffusion assay according to the guidelines from the Clinical and Laboratory Standards Institute [53]. Briefly, each of the *S. aureus* strains was aerobically grown at 37 °C on the BHI agar plate for 18 h, and colonies were picked and suspended in sterilized saline to meet turbidity of 0.5 McFarland standard. Each bacterial suspension was spread on the Mueller–Hinton agar plate and incubated in the presence of antibiotic discs at 35 °C for 18 h. The antibiotic discs used were cefoxitin (30 μg), oxacillin (1 μg), penicillin (10 μg), erythromycin (15 μg), clindamycin (2 μg), and chloramphenicol (30 μg) purchased from Thermo Fisher Scientific (Waltham, MA, USA). After incubation, the diameters of the inhibition zones were measured to evaluate antibiotic resistance or susceptibility of *S. aureus* strains.

### 4.3. Collection of Putative Endolysins against MRSA

Complete *S*. *aureus* genome sequences were retrieved from the National Center for Biotechnology Information (NCBI; retrieval date: 6 February 2020) and annotated using a Rapid Annotation using Subsystems Technology server [70]. After annotation, *S*. *aureus* strains containing PBP2a were considered MRSA [23], and putative endolysin sequences were retrieved from their genomes.

### 4.4. Conserved Domain Analysis, Sequence Alignment, and Solubility Prediction of Putative Endolysins

The conserved domains of putative endolysins were analyzed using the NCBI conserved domain database [71], and the distribution of the conserved domains was visualized using the ggplot2 package [72] in R software (version 4.1.3) [73]. Putative endolysins containing both an EAD and a CBD were selected. Multisequence alignments based on amino acid sequences were determined via multiple alignments using fast Fourier-transform (MAFFT v7.505) with the E-INS-i algorithm [74]. Alignments were used to create phylogenetic reconstructions of putative endolysins using FastTree [75] and default parameters (amino acid distances BLOSUM45 and the Jones–Taylor–Thorton model), and the resulting phylogenetic tree was visualized using ggtree [76] in R software (version 4.1.3) [73]. For all putative endolysin sequences with a CBD, protein solubility was predicted using various prediction tools, including SoluProt [49], Protein-sol [48], and SKADE [50]. The resulting solubility values were then transformed into Z scores, and endolysin candidates were selected based on the highest two Z-score sums.

### 4.5. Identification, Cloning, and Overexpression of Endolysin LyJH1892

Based on the solubility index, putative endolysins 177 and 117 were selected as endolysin candidates and named LyJH1508 and LyJH1892, respectively. The sequences of LyJH1508 and LyJH1892 were acquired from the whole genome sequence of *S*. *aureus* (GenBank accession numbers: GCA_003193885.1 and GCA_000568455.1, respectively). Both endolysins were chemically synthesized after codon optimization to facilitate overexpression of the recombinant protein in *E*. *coli* (BIONICS Co., Ltd., Seoul, Republic of Korea), and the chemically synthesized endolysin genes were amplified by polymerase chain reaction (PCR) using HiPi™ plus thermostable Taq DNA polymerase (Elpis-biotech, Daejeon, Republic of Korea). Purified DNA fragments were digested using the restriction enzymes *Bam*H1 and *Xho*1 (New England Biolabs, Ipswich, MA, USA) and then cloned into the expression vector pET28b (Novagen Inc., Madison, WI, USA) with an N-terminal hexa histidine-tag sequence. The cloned pET28b vector was transformed into competent *E*. *coli* BL21 (DE3) cells. The transformants were cultured in LB broth until the optical density at 595 nm (OD_595nm_) reached 0.4, and 0.5 mM isopropyl-β-D-thiogalactopyranoside was added to the medium to overexpress the target recombinant endolysin. Cells were further incubated for 18 h at 16 °C. Harvested cells were resuspended in lysis buffer (50 mM NaH_2_PO_4_, 300 mM NaCl, 10 mM imidazole, pH 8.0), and the resuspended cells were lysed on ice using a sonicator (KYY-80, Korea Process Technology Co., Ltd., Seoul, Republic of Korea). After centrifugation at 12,000× *g* for 15 min, the crude extracts were purified using Nuvia™ IMAC resin charged with Ni (Bio-Rad Laboratories, Inc., Hercules, CA, USA), according to manufacturer instructions. Purified samples were subjected to SDS-PAGE analysis. The purified endolysin was pooled and dialyzed against the elution buffer (50 mM NaH_2_PO_4_, 300 mM NaCl, and pH 8.0).

### 4.6. Structure Prediction of LyJH1892

To predict the 3D structure of LyJH1892, we used Colab Fold notebook, which predicts protein structure using AlphaFold2 combined with a fast multiple sequence alignment generation stage via MMseq2 [77]. The predicted structure was visualized using ChimeraX 1.3 [78].

### 4.7. Characterization and Lytic Spectrum of LyJH1892

*Staphylococcus aureus* (NCCP 16830) was used as the reference strain for the lytic test of LyJH1892. Cells of this strain were cultured to an OD_595nm_ of 0.8–1.0 and then harvested. To characterize the lytic activity of the endolysin, 20 μL of LyJH1892 (100 μg/mL) was added to 96-well plates (SPL Life Sciences Co., Ltd., Pocheon, Republic of Korea) containing cell suspensions (180 μL) and subjected to various conditions (pH, temperature, NaCl concentration, and metal ions). For control samples, 20 μL of elution buffer was added instead of endolysin. The OD_595nm_ values were monitored using an iMark microplate reader (Bio-Rad Laboratories). The lytic activity of LyJH1892 was calculated after 2 h as follows: 100 × (∆OD_595_ after LyJH1892 treatment − ∆OD_595_ treated with elution buffer)/initial OD_595_. To determine the optimal pH range for endolysin activity, the indicator strain was resuspended in 50 mM sodium phosphate (pH 6.0, 7.0, and 8.0) and sodium glycine buffer (pH 9.0, 10.0) and incubated at 25 °C for 120 min. To determine the optimal temperature for LyJH1892, samples were incubated at different temperatures (4, 16, 25, 37, 50, and 60 °C) for 120 min. The effect of NaCl on LyJH1892 activity was evaluated by adding 0, 31.3, 62.5, 125, 250, and 500 mM NaCl to the empirically determined pH buffer. The effect of divalent cations on LyJH1892 was measured as described by Kim et al. [20]. Briefly, LyJH1892 (100 μg/mL) was incubated with 5 mM EDTA at 25 °C for 30 min to chelate divalent cations attached to the endolysin, and then EDTA was removed by changing the elution buffer to Amicon Ultra-4 (10 kDa) (Merck KGaA, Darmstadt, Germany) [79]. LyJH1892 was incubated with 10 mM CaCl_2_, MgCl_2_, MnCl_2_, or ZnCl_2_ at 25 °C for 30 min, and its lytic activity was assessed. To determine the dose-dependent response of LyJH1892 under optimal conditions, the indicator strain was resuspended in an empirically determined buffer (sodium glycine buffer, pH 9.0, and 0 mM NaCl), and then serially diluted LyJH1892 (20 μL, 3.125–100 μg/mL concentration) or elution buffer (20 μL, for control) was added to the suspension. The mixture was then incubated at 25 °C, and the OD_595nm_ values were monitored at 0, 10, 30, 60, 90, and 120 min. A list of the bacterial strains used in the lytic spectrum test is presented in Table 3. All bacterial species used in the lytic spectrum test were grown as described above to an OD_595nm_ of 0.8–1.0 before being harvested and resuspended in the empirically determined buffer. The lytic activity of LyJH1892 was evaluated after incubation for 2 h at 25 °C. All experiments were performed in triplicate.

### 4.8. Statistical Analysis

Statistical analysis was performed using R software (version 4.1.3) [73]. Data derived from the characterization of the LyJH1892 lytic activity in triplicate were analyzed using the non-parametric Kruskal–Wallis test using the Kruskal–Wallis test function because the residuals did not meet the normal distribution. A post hoc Dunn test (Dunn Test function in the FSA package) [80] was used to compare differences among treatments if a significant difference was observed. All *p*-values were adjusted using the Benjamini–Hochberg false discovery rate, and statistical significance was set at *p* < 0.05.

## 5. Conclusions

In summary, we present a novel strategy for identifying new endolysin candidates against MRSA from prophage sources by combining bioinformatics approaches. In total, 272 putative endolysin candidates were detected from the MRSA genome sequences, and only 114 putative endolysins exhibited multi-conserved domain combinations possessing EAD and CBD domains. In addition, we successfully developed a new endolysin, LyJH1892, with broad lytic activity against MSSA, MRSA, and several CNSs; LyJH1892 may represent a useful biocontrol agent against bovine mastitis. This study provides a rapid and useful strategy for the development of specific endolysins against antibiotic-resistant bacterial strains.

## Figures and Tables

**Figure 1 ijms-24-05772-f001:**
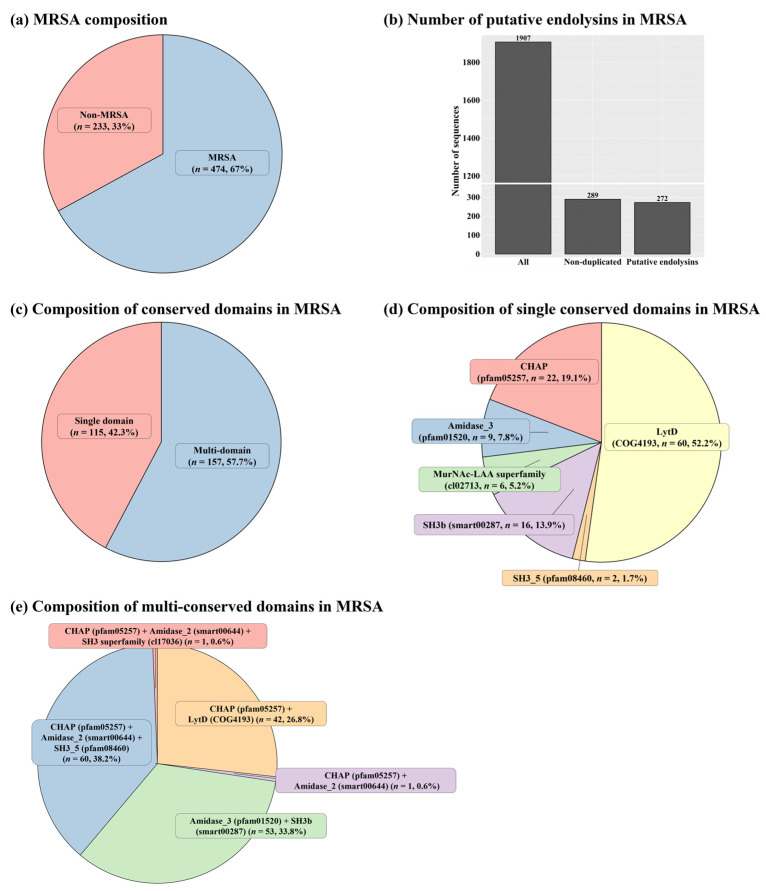
General information regarding *Staphylococcus aureus* genomes and putative endolysins. (**a**) Distribution of *S. aureus* and methicillin-resistant *S. aureus* (MRSA). (**b**) Quantity of putative endolysins observed in the MRSA genomes. (**c**) Distribution of domain types (single versus multiple domains). (**d**) Types of domains observed in single-domain putative endolysins. (**e**) Types of domain combinations observed in multiple-domain putative endolysins. The conserved domains were analyzed using the database from the National Center for Biotechnology Information.

**Figure 2 ijms-24-05772-f002:**
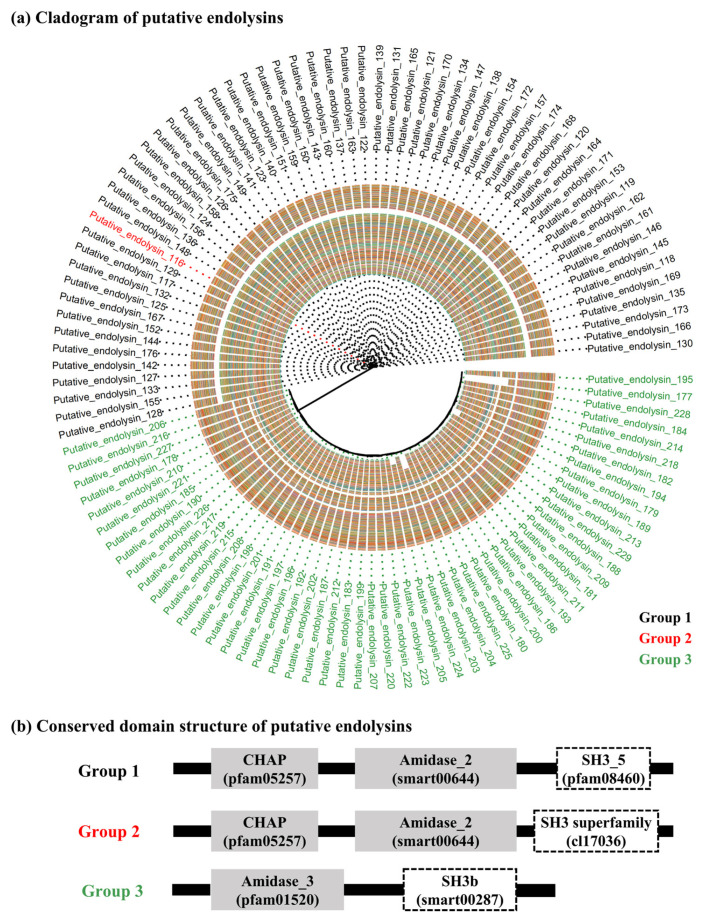
Diversity of putative endolysins with both enzymatically active domains and cell wall binding domains identified from methicillin-resistant *Staphylococcus aureus* related genomes. (**a**) Cladogram of putative endolysins. In total, 114 putative endolysins were multi-aligned based on the amino acid sequences using the E-INS-I algorithm on multiple alignments using fast Fourier transform (version 7.505). The resulting alignments were phylogenetically reconstructed using FastTree, and the cladogram was constructed using ggtree in R software (version 4.1.3). Color tiles in the cladogram represent multi-alignment results among 114 putative endolysins. (**b**) Conserved domain structure of putative endolysins. The gray boxes represent enzymatically active domains and the white boxes show cell wall binding domains. CHAP, cysteine, histidine-dependent amidohydrolases/peptidases.

**Figure 3 ijms-24-05772-f003:**
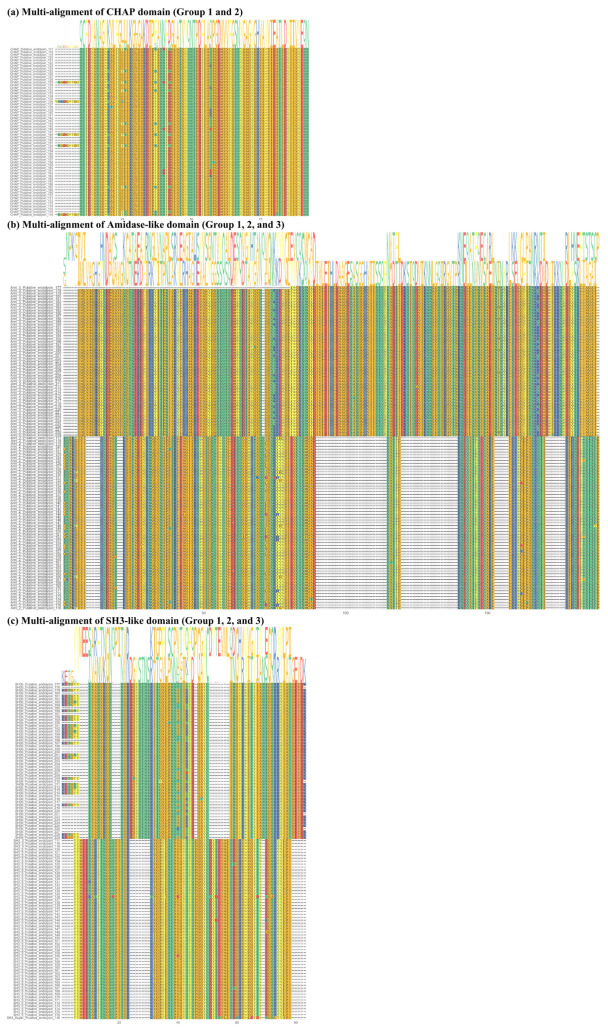
Multi-alignment of all putative endolysins with an enzymatically active domain and a cell wall binding domain based on amino acid sequences. (**a**) Multi-alignment of the cysteine, histidine-dependent amidohydrolases/peptidases domain from groups 1 and 2 (Accession: pfam05257 and sequence identity: 82.4%). (**b**) Multi-alignment of the Amidase-like domains (Accession: Ami_2 (smart00644) and Amidase_3 (pfam01520); sequence identity: 12.7%). (**c**) Multi-alignment of the SH3-like domain (Accession: SH3_5 (pfam08460), SH3 superfamily (cl17036), and SH3b (smart00287); sequence identity: 22.9%).

**Figure 4 ijms-24-05772-f004:**
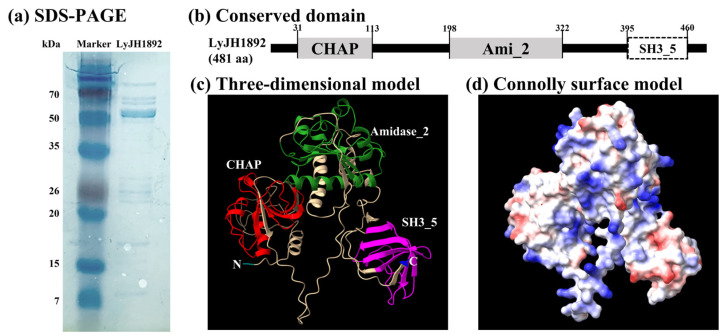
Structural characteristics of endolysin LyJH1892. (**a**) Analysis of endolysin LyJH1892 by sodium dodecyl sulfate-polyacrylamide gel electrophoresis. Lane 1 contained a protein molecular weight marker and lane 2 contained purified LyJH1892. (**b**) The conserved domain of LyJH1892. The gray square represents the enzymatically active domain, and the white square denotes the cell wall binding domain. (**c**) Three-dimensional model of LyJH1892, as predicted by AlphaFold2 using the Colab Fold notebook. (**d**) Connolly surface form of LyJH1892 created in ChimeraX 1.3. Blue and red colors represent the most positive and most negative polar activities, respectively.

**Figure 5 ijms-24-05772-f005:**
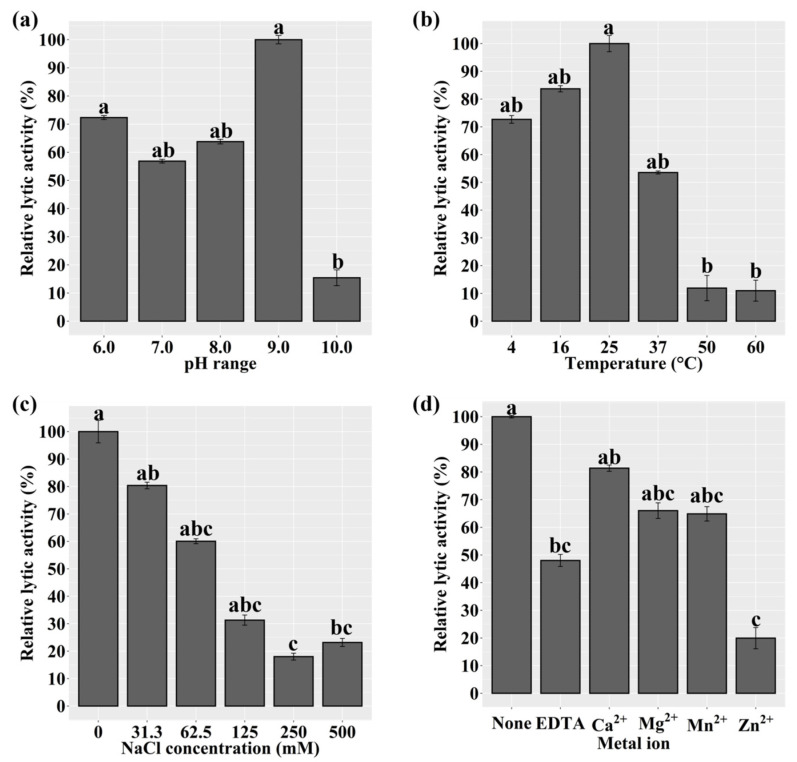
Lytic activity of LyJH1892 against *Staphylococcus aureus* (NCCP 16830) at various (**a**) pH levels, (**b**) temperatures, (**c**) NaCl concentrations, and (**d**) metal ion additions. Data are shown as means ± standard deviation of triplicate assays. In Figure 5d, None, pure LyJH1892; EDTA, purified LyJH1892 after incubating with 5 mM ethylenediaminetetraacetic acid (EDTA); Metal ion (Ca^2+^, Mg^2+^, Mn^2+^, or Zn^2+^), LyJH1892 after adding 10 mM metal ions to EDTA-treated LyJH1892. Bars with different superscript letters (a–c) indicate a significant difference (*p* < 0.05).

**Figure 6 ijms-24-05772-f006:**
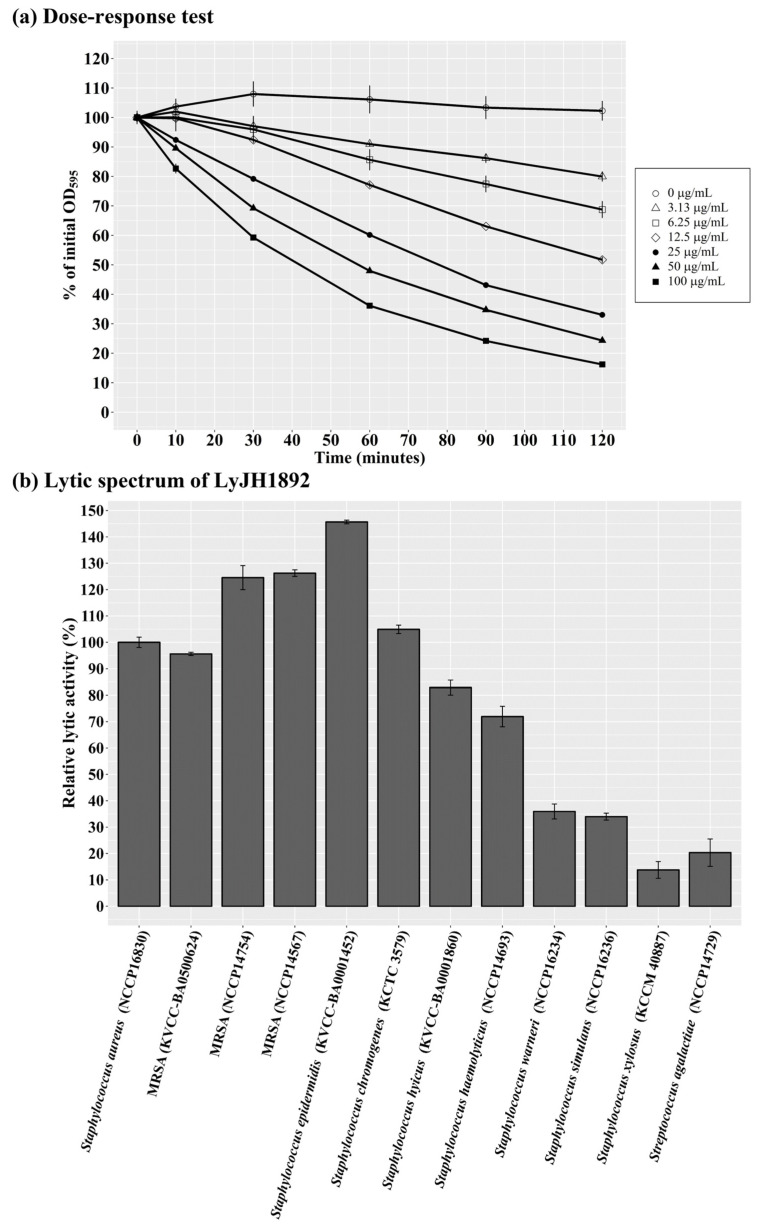
Optimal lytic activity and lytic spectrum of LyJH1892. (**a**) Dose–response test against *Staphylococcus aureus* (NCCP 16830). (**b**) A lytic spectrum of LyJH1892. All lytic tests were performed under optimal conditions of LyJH1892 (pH 9.0, 25 °C, and no addition of NaCl and metal ions). Data are shown as means ± standard deviation of triplicate assays.

**Table 1 ijms-24-05772-t001:** Top 10 of predicted recombinant protein solubility of putative endolysins based on Soluprot, Protein-Sol, and SKADE ^1^.

Group ^2^	Name	Amino Acids	Soluprot		Protein-Sol		SKADE		Sum ^7^
			Solubility ^3^	Z-Score ^4^	Scaled-sol ^5^	Z-Score ^4^	Solubility ^6^	Z-Score ^4^	
Group 3	Putative endolysin 177	249	0.569	−0.269	0.680	5.559	0.186	−1.638	3.652
Group 1	Putative endolysin 117	481	0.735	1.114	0.354	−0.276	0.327	0.903	1.741
Group 1	Putative endolysin 129	481	0.776	1.465	0.343	−0.473	0.311	0.610	1.601
Group 1	Putative endolysin 121	481	0.749	1.239	0.332	−0.670	0.330	0.951	1.520
Group 1	Putative endolysin 159	481	0.718	0.973	0.332	−0.670	0.343	1.182	1.485
Group 1	Putative endolysin 132	481	0.750	1.241	0.343	−0.473	0.312	0.631	1.398
Group 1	Putative endolysin 147	481	0.727	1.052	0.332	−0.670	0.332	0.981	1.364
Group 1	Putative endolysin 143	481	0.741	1.167	0.322	−0.849	0.334	1.015	1.333
Group 1	Putative endolysin 174	481	0.722	1.006	0.332	−0.670	0.332	0.986	1.322
Group 1	Putative endolysin 150	481	0.732	1.092	0.322	−0.849	0.336	1.063	1.306

^1^ The top 10 114 putative endolysins by predicted solubility are presented in this table, and the complete dataset for predicted solubility is presented in Appendix A. ^2^ Group 1, CHAP domain + Ami_2 domain + SH3_5 domain; group 2, CHAP domain + Ami_2 domain + SH3 superfamily domain; group 3, Amidase_3 domain + SH3b domain. ^3^ Solubility, predicted solubility scores based on Soluprot. ^4^ Z-score = (x − μ)/σ, where x is the raw score, μ is the mean, and σ is the standard deviation. ^5^ Scaled-sol, predicted solubility scores based on Protein-Sol. ^6^ Solubility, predicted solubility scores based on SKADE. ^7^ Sum, the sum of Z-scores calculated from Soluprot, Protein-Sol, and SKADE.

**Table 2 ijms-24-05772-t002:** Antibiotic resistance profiles of *Staphylococcus aureus* strains.

Antibiotic Disc	Antibiotic Susceptibility ^1^			
	NCCP 16830	NCCP 1457	NCCP 14754	KVCC-BA0500624
Cefoxitin (30 μg)	Susceptible	Resistant	Resistant	Resistant
Oxacillin (1 μg)	Susceptible	Resistant	Resistant	Resistant
Penicillin (10 μg)	Resistant	Resistant	Resistant	Resistant
Erythromycin (15 μg)	Susceptible	Resistant	Resistant	Resistant
Clindamycin (2 μg)	Susceptible	Susceptible	Intermediate	Susceptible
Chloramphenicol (30 μg)	Susceptible	Susceptible	Susceptible	Susceptible

^1^ Antibiotic susceptibility was evaluated by measuring the diameter of the clear zone.

**Table 3 ijms-24-05772-t003:** Bacterial strains and growth conditions.

Bacterial Strain	Purpose	Growth Condition ^1^
*Escherichia coli* DH5α	Cloning host	LB broth
*E. coli* BL21(DE3)	Expression host	LB broth
*Staphylococcus aureus* (NCCP 16830)	Indicator strain	BHI broth
MRSA (KVCC-BA0500624)	Lytic spectrum	BHI broth
MRSA (NCCP 14567)	Lytic spectrum	BHI broth
MRSA (NCCP 14754)	Lytic spectrum	BHI broth
*Staphylococcus hyicus* (KVCC-BA0001860)	Lytic spectrum	BHI broth
*Staphylococcus epidermidis* (KVCC-BA0001452)	Lytic spectrum	BHI broth
*Staphylococcus haemolyticus* (NCCP 14693)	Lytic spectrum	BHI broth
*Staphylococcus simulans* (NCCP 16236)	Lytic spectrum	BHI broth
*Staphylococcus warneri* (NCCP 16234)	Lytic spectrum	BHI broth
*Staphylococcus chromogenes* (KCTC 3579)	Lytic spectrum	BHI broth
*Staphylococcus xylosus* (KCCM 40887)	Lytic spectrum	BHI broth
*Streptococcus agalactiae* (NCCP 14729)	Lytic spectrum	BHI broth

MRSA, methicillin-resistant *Staphylococcus aureus*; BHI, brain heart infusion; LB, Luria-Bertani. ^1^ All bacterial strains used in this study were grown aerobically at 37 °C.

## Data Availability

The data presented in this study are available on request from the corresponding author.

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
