# Peer review of "A Novel Strategy to Identify Endolysins with Lytic Activity against Methicillin-Resistant *Staphylococcus aureus"

_ijms, 2023, doi:10.3390/ijms24065772_

Round 1

Reviewer 1 Report

Journal                 IJMS (ISSN 1422-0067)

Manuscript ID     ijms-2221519

Type                    Article

Title     A novel strategy to identify endolysin with lytic activity against methicillin-resistant Staphylococcus aureus

Authors   Hanbeen Kim , Jakyeom Seo *

Section   Molecular Microbiology

Special Issue Bacteriophage Biology: From Genomics to Therapy

Comments

 The strategy for finding effective endolysins is well executed.

A few queries are as follows:

What is the basis of the specificity against MRSA?

Can this endolysin be used against MSSA?

Can the strategy be extended to act against Gram-Negative bacteria as well?

Are endolysins better compared to quorum sensing inhibitors as alternatives to antibiotics?

Author Response

Response to the Reviewer 1

[Response] Thank you for your scientific and invaluable review and comments for our manuscript. We have incorporated all the comments and suggestions, and also we have received English revisions. The changes are marked using the “tract changes function” for a more convenient review process. Here is a point-by-point response to the reviewer’s comments and concerns.

1. What is the basis of the specificity against MRSA?

[Response] Endolysins are bacteriophage-derived peptidoglycan hydrolases and are utilized in the final stage of the bacteriophage replication cycle to release newly produced progeny from the host bacterial cell by breaking down the peptidoglycan layer, thereby inducing the lysis of host bacteria. Endolysins active against gram-positive bacteria have generally shown the N-terminal enzymatically active domain (EAD) and the C-terminal cell wall-binding domain (CBD), and a CBD of endolysin determines the spectrum of lytic activity by recognizing and binding to the peptidoglycan layer. Therefore, we hypothesized the endolysin having a CBD identified from MRSA genomes might have lytic activity against MRSA. In addition, we have revised and added the basis of the specificity against MRSA in the introduction (Lines 48-58 and 75-76).

2. Can this endolysin be used against MSSA?

[Response] We have added an antibiotic susceptibility test for S. aureus strains used in this study (Line 219-227). Among the four strains of S. aureus, S. aureus (NCCP 16830) was determined to be MSSA and the others were evaluated as MRSA. Considering the result from the lytic spectrum test, LyJH1892 can be used for controlling both MSSA and MRSA.

3. Can the strategy be extended to act against Gram-Negative bacteria as well?

[Response] We think our strategy can be used to develop endolysins target Gram-negative bacteria, but, honestly, several modifications are needed. In general, endolysins that target Gram-negative bacteria consist of only an enzymatically active domain, which makes the development of endolysins against Gram-negative bacteria difficult. Therefore, several studies have tried to solve this problem (E.g., adding a polycationic nonapeptide into endolysin sequences). Literally, we are trying to find bacteriophage tail protein sources and we will test the effectiveness of combining tail protein and endolysin sequences against Gram-negative bacteria. If this strategy is effective, then we will extend our strategy to Gram-negative bacteria.

4. Are endolysins better compared to quorum sensing inhibitors as alternatives to antibiotics?

[Response] As far as we know, the lytic or killing spectrum of endolysins is more specific than that of quorum-sensing inhibitors. Given that we have not conducted direct tests on the impact of quorum sensing and endolysin on target microorganisms, it is challenging to determine which approach would be more effective. We think that the specific lytic activity against pathogens is one of the most important factors in this field, thus we think the use of endolysins might be better than that of quorum-sensing inhibitors.

Reviewer 2 Report

This study has been very well characterized the novel endolysin, specifically sequence analysis, that can inhibit MRSA. There are few minor comments.

1.     Antibiotic susceptibilities of strains used in this study, including MRSA, should be included with their antibiotic resistance profiles, obtained from broth dilution and disc diffusion assays.

2.     The source of endolysin (from prophage) should be stated in more detail.

3.     Since this study highlights the effectiveness of endolysin as alternative over antibiotic and phages themselves, the lytic (inhibitory) activities of antibiotic and phages against MRSA should be compared to those of endolysin.

4.     More discussion is needed for the result that pH 9 shows most stable and high lytic activity, even more than pHs 6 and 7.

Author Response

Response to the Reviewer 2

[Response] Thank you for your scientific and invaluable review and comments on our manuscript. We have incorporated all the comments and suggestions, and also we have received English revisions from Editage. The changes are marked using the “tract changes function” for a more convenient review process.

Here is a point-by-point response to the reviewer’s comments and concerns.

1. Antibiotic susceptibilities of strains used in this study, including MRSA, should be included with their antibiotic resistance profiles, obtained from broth dilution and disc diffusion assays.

[Response] Thank you for your scientific ask. We have added the antibiotic susceptibility test for S. aureus strains against several antibiotics including cefoxitin (30 μg), oxacillin (1 μg), penicillin (10 μg), erythromycin (15 μg), clindamycin (2 μg), and chloramphenicol (30 μg), and added the method and result in the manuscript (Line 219-227, 464-475, and Figure S3). 

2. The source of endolysin (from prophage) should be stated in more detail.[Response] We have represented the source of endolysins in Table S2 and lines 502-504.

3. Since this study highlights the effectiveness of endolysin as alternative over antibiotic and phages themselves, the lytic (inhibitory) activities of antibiotic and phages against MRSA should be compared to those of endolysin.

[Response] Thank you for your critical mention. We have added more information about your mention in lines 399-416.

4. More discussion is needed for the result that pH 9 shows most stable and high lytic activity, even more than pHs 6 and 7.

[Response] To be honest, we don’t know why the LyJH1892 showed the most stable and high lytic activity on pH 9, rather than pH 6 or 7. We have also tried to find bioinformatic tools to predict optimal pH based on sequence, but we can’t find any tools so far. If you don’t mind we would like to learn your opinion about this unexpected optimal pH.

Round 2

Reviewer 2 Report

This has been well revised based on the review's comments. But, technical and biological replications should be stated in the Statistical analysis section. And, specifically, the pH effect on lytic activity should be further confirmed whether the results are reproduciable. 

Author Response

Thank you for your kind and scientific response.

Actually, we tested all characterizations of LyJH1892 in triplicate (technically and biologically). We have added the information on replications in the Statistical analysis section as you mentioned. 

Thank you very much for taking the time to review the paper.